

# Ubiquitin specific peptidases and prostate cancer

Yunfei Guo[1], Shuaishuai Cui[1], Yuanyuan Chen[1], Song Guo[1] and Dahu Chen[1]

Shandong University of Technology, School of Life Sciences and Medicine, Zibo, Shandong, China

## ABSTRACT

Protein ubiquitination is an important post-translational modification mechanism, which regulates protein stability and activity. The ubiquitination of proteins can be reversed by deubiquitinating enzymes (DUBs). Ubiquitin-specific proteases (USPs), the largest DUB subfamily, can regulate cellular functions by removing ubiquitin(s) from the target proteins. Prostate cancer (PCa) is the second leading type of cancer and the most common cause of cancer-related deaths in men worldwide. Numerous studies have demonstrated that the development of PCa is highly correlated with USPs. The expression of USPs is either high or low in PCa cells, thereby regulating the downstream signaling pathways and causing the development or suppression of PCa. This review summarized the functional roles of USPs in the development PCa and explored their potential applications as therapeutic targets for PCa.

## INTRODUCTION

Regulation of gene expression is a complex process, involving interactions among DNA, RNA, proteins, and environment. The regulatory process of gene expression occurs at multiple levels to maintain the optimal levels of RNAs and proteins (*Tran & Hutvagner, 2013*). As an important regulatory mechanism, post-translational modifications (PTMs) are essential for cellular activities. There are several types of PTMs, such as phosphorylation, acetylation, methylation, sumoylation, and ubiquitination; each of them corresponds to one or more specific functions (*Millar et al., 2019*).

The ubiquitination of proteins plays a very important role in regulating protein function (*Komander & Rape, 2012*). It can regulate almost all aspects of cellular physiology (*Hu et al., 2021*). Ubiquitin, discovered in the 1980s (*Cappadocia & Lima, 2018*), is an 8.5-kDa small protein, containing 76 amino acids. This protein can be post-translationally attached to the substrate protein, usually at a lysine residue (*Clague, Heride & Urbé, 2015*; *Rennie, Chaugule & Walden, 2020*). As shown in Fig. 1, ubiquitin can covalently conjugate to the substrate proteins by three different enzymes, including E1, E2, and E3, during ubiquitination (*Clague, Heride & Urbé, 2015*). In eukaryotes, the ubiquitin-proteasome system, which is mainly composed of ubiquitin molecules, ubiquitylating enzymes, and 26S proteasomes, is an important pathway for the degradation of damaged or misfolded proteins (*Chauhan et al., 2021*; *Svikle et al., 2022*).

Corresponding author
Dahu Chen, dahuchen@sdut.edu.cn, dahuchen@outlook.com

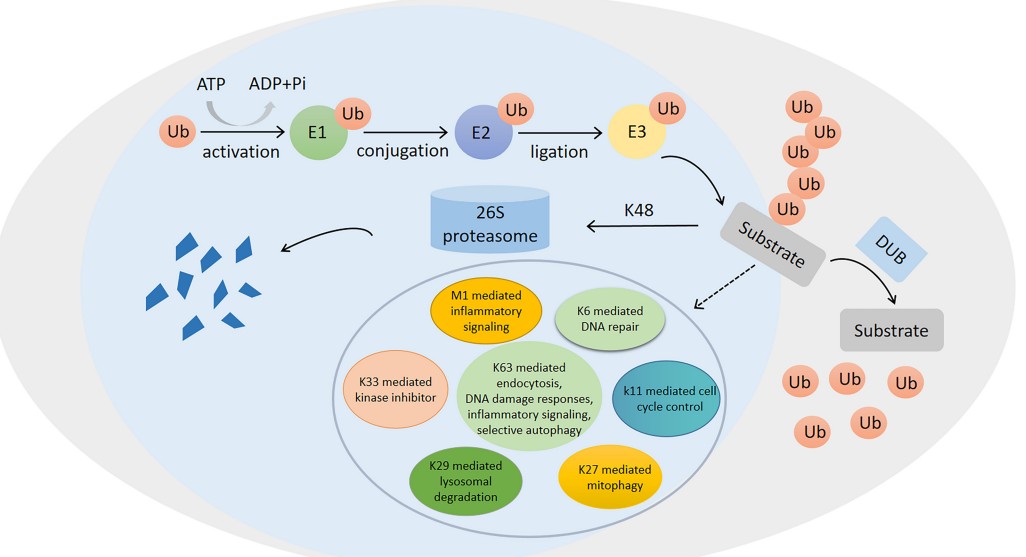

**Figure 1** **Ubiquitination/deubiquitination modifications and their regulatory role in protein degradation and cellular physiological functions.** The process of ubiquitin (ub) binding to substrate proteins in the presence of E1, E2 and E3. This process is reversible and can remove ubiquitination modifications of substrate proteins in the presence of deubiquitinating enzymes (DUB). ubiquitination modifications at position K48 are proteins that undergo degradation to protein fragmentation *via* the 26S proteasome, while ubiquitination modifications at positions K6, K11, K27, K29, K33, K63 and M1 are involved in the regulation of numerous physiological processes in the cell.

Ubiquitination is a covalent modification, which can be reversed by deubiquitinating enzymes (DUBs) (*Bhoj & Chen, 2009*; *Chen, Liu & Zhou, 2021a*). DUBs are proteases, which cleave peptide or isopeptide bonds between the two linked ubiquitin molecules or between ubiquitin and modified proteins (*Clague, Urbé & Komander, 2019*). It can reverse the E3 ubiquitin ligase-mediated protein modifications (*Cheng et al., 2019b*), thereby protecting proteins from degradation by the proteasomes (*Haq, Suresh & Ramakrishna, 2018*). Similar to ubiquitination, deubiquitination, catalyzed by DUBs, also plays a critical role in regulating cell functions (*Schmidt et al., 2005*; *Daniel & Grant, 2007*; *Kennedy & D'Andrea, 2005*; *Lai, Chen & Tse, 2020*; *Ramakrishna, Suresh & Baek, 2011*; *Ramakrishna, Suresh & Baek, 2015*). Recently, numerous DUBs have shown correlations with many diseases, such as cancer, inflammatory diseases, diabetes, and neurodegenerative diseases (*Chen, Liu & Zhou, 2021a*; *Leznicki & Kulathu, 2017*). The human genome encodes more than 100 DUBs, most of which are cysteine proteases (*Leznicki & Kulathu, 2017*). DUBs are divided into six families, including ubiquitin-specific proteases (USPs), ubiquitin C-terminal hydrolases (UCHs), ovarian tumor-related proteases (OTUs), Machado-Joseph disease protein domain proteases (MJDs), motif interacting with Ub-containing novel DUB family (MINDYs), and zinc finger-containing ubiquitin peptidase 1 (ZUP1) (*Harrigan et al., 2018*; *Kwasna et al., 2018*; *Lei et al., 2021*). USPs are the largest subfamily of DUBs, which are closely correlated with the development and progression of numerous cancer types, such as colorectal cancer (CRC) (*Li et al., 2020*), prostate cancer (PCa) (*Liao et al.,*

*2021a*), liver cancer (*Du et al., 2021*), and breast cancer (*Ma et al., 2019*). Moreover, several USPs can also inhibit the occurrence and progression of cancer, such as glioblastoma (*Kit Leng Lui et al., 2017*).

PCa is the second leading type of cancer and the most common cause of cancer-related deaths in men worldwide (*Vietri et al., 2021*). The therapeutic strategies for PCa have been improved, albeit modestly, in the past few years, increasing the overall survival times of PCa patients (*Nag & Dutta, 2020*). PCa is highly dependent on androgen receptor (AR) signaling. AR is important for the growth and differentiation of normal prostate cells and plays a key role in the pathogenesis of PCa (*Smolle et al., 2017*). Therefore, androgen deprivation therapy (ADT) is the main treatment option for metastatic PCa (mPCa) (*Sena & Denmeade, 2021*; *Tsujino et al., 2021*). Despite its effectiveness, the patients receiving ADT can develop resistance, eventually developing castration-resistant PCa (CRPCa) (*Fujita & Nonomura, 2019*). Abiraterone and enzalutamide are second-generation ADTs, which are used for the treatment of CRPCa; however, cancer also eventually progresses to metastatic CRPCa (mCRPCa) (*Berish et al., 2018*). In addition to AR, the loss of tumor suppressors and aberrant activation or expression of oncogenes also play a crucial role in the progression of PCa. For example, the tumor suppressor genes *p53* and *PTEN* and oncogene *Myc* are involved in the development and progression of PCa (*Logothetis et al., 2013*; *Whitlock et al., 2020*). In addition, numerous studies have also shown a strong correlation between the expression of DUBs and the progression of PCa (*Deng et al., 2021*; *Ge et al., 2021*; *Jin et al., 2018*; *McClurg & Robson, 2015*; *Peng et al., 2022*).

Ubiquitination and deubiquitination can regulate various aspects of human cell biology and physiology. The malfunction of these modifications can cause numerous diseases, including cancers. There has been a growing interest in exploiting the components of ubiquitination and deubiquitination machinery as therapeutic targets. Due to their greater numbers and diversity, DUBs are intrinsically attractive as potential drug targets. More than half of the total DUBs are USPs, which are cysteine peptidases. Currently, USPs have been extensively studied in various types of cancer. In terms of USPs' role in PCa, *Islam et al. (2019)* showed that targeted ubiquitinase could regulate the therapeutic pathways of PCa. This review article summarized the expression patterns of different USPs in PCa and correlations between these expression patterns and the overall survival times of PCa patients as well as highlighted the mechanism of USPs, regulating the development and progression of PCa. This review article might provide basic knowledge for general readers as well as cancer researchers, especially those who are dedicated to PCa research and developing PCa drug targets.

## SURVEY METHODOLOGY

PubMed database was used for related literature search using the keyword "USP," "ubiquitin specific peptidase," "ubiquitin-proteasome system," "ubiquitination," "deubiquitination," "prostate cancer," and "cancer."

## Structural characteristics of USPs

USPs were first discovered and cloned in *Saccharomyces cerevisiae* (*Tobias & Varshavsky, 1991*). They are the largest subfamily of DUBs having a total of 58 members. The number of USPs has increased since the evolution of E3 ubiquitin ligases (*Semple, 2003*). The size of USPs ranges from 330 amino acids to 3,500 amino acids with an average size of 800–1,000 amino acids for full-length enzymes. The catalytic structural domain of USPs contains 295-850 amino acids; the catalytic structural domain of 27 USPs contains 300–400 amino acids, while that of 29 USPs contains 400–850 amino acids (*Ye et al., 2009*). USPs have also other diverse domains in terms of size and structure (*Hariri & St-Arnaud, 2021*). However, there is a high degree of homology within the catalytic domain. The catalytic core of USPs contains three motifs, consisting of very conserved catalytic Cys residues, catalytic His residues, and catalytic Asp/Asn residues, which form the catalytic triad (*Nijman et al., 2005*; *Ye et al., 2009*). In addition to the catalytic domain, USPs also have domains for subcellular localization, substrate specificity, zinc binding, and ubiquitin recognition (*Hariri & St-Arnaud, 2021*; *Nijman et al., 2005*; *Ye et al., 2009*). Figure 2A shows USP4, USP7, USP14, USP19, and USP44 as examples of USPs having major domains.

The ubiquitin-like (UBL) domain of USPs can regulate their catalytic activity; however, the mechanism of action of the UBL domain in each USP varies. For example, the UBL domain of USP14 is important for its localization on proteasome and might enhance its catalytic capability., while that of USP4 binds to the catalytic domain, showing a competitive relationship with ubiquitin. As shown in Fig. 2B, the UBL4 and UBL5 domains of USP7 are located on its C-terminal and can affect its deubiquitinating activity by promoting conformational changes and facilitating the formation of a catalytic center (*Faesen, Luna-Vargas & Sixma, 2012*).

Different USPs have specific substrate proteins; therefore, they can regulate different signaling pathways (*Ye et al., 2009*). USPs can stabilize various oncoproteins or alter their cellular localization by deubiquitination, which can cause the development and progression of cancer (*Chauhan et al., 2021*). Numerous studies have shown that targeting USPs might be a promising therapeutic approach for cancer treatment (*Dai et al., 2020*; *Du et al., 2021*; *Li et al., 2020*; *Ma et al., 2019*; *Nininahazwe et al., 2021*; *Zhu et al., 2020*).

## USPs and PCa
### USP1

The mRNA expression levels of *USP1* were higher in the PCa tissues as compared to those in the normal tissues. However, there were no differences in protein expression levels between PCa and normal tissues. USP1 could promote the proliferation of PCa cells, while the knockdown of the *USP1* gene led to the inhibition of cellular proliferation (*Liao et al., 2021a*). Therefore, USP1 might play a promotional role in the development of PCa. Moreover, USP1 was highly expressed in cancer and was strongly correlated with the poor prognosis of patients with numerous cancer types, such as liver cancer, breast cancer, and multiple myeloma (*Das et al., 2017*; *Liao et al., 2021b*; *Ma et al., 2019*). These studies further supported USP1 as a potential target for cancer therapy. USP1 activity can be regulated by binding to UAF1 (USP1 associated factor 1), a protein, which contains a WD40

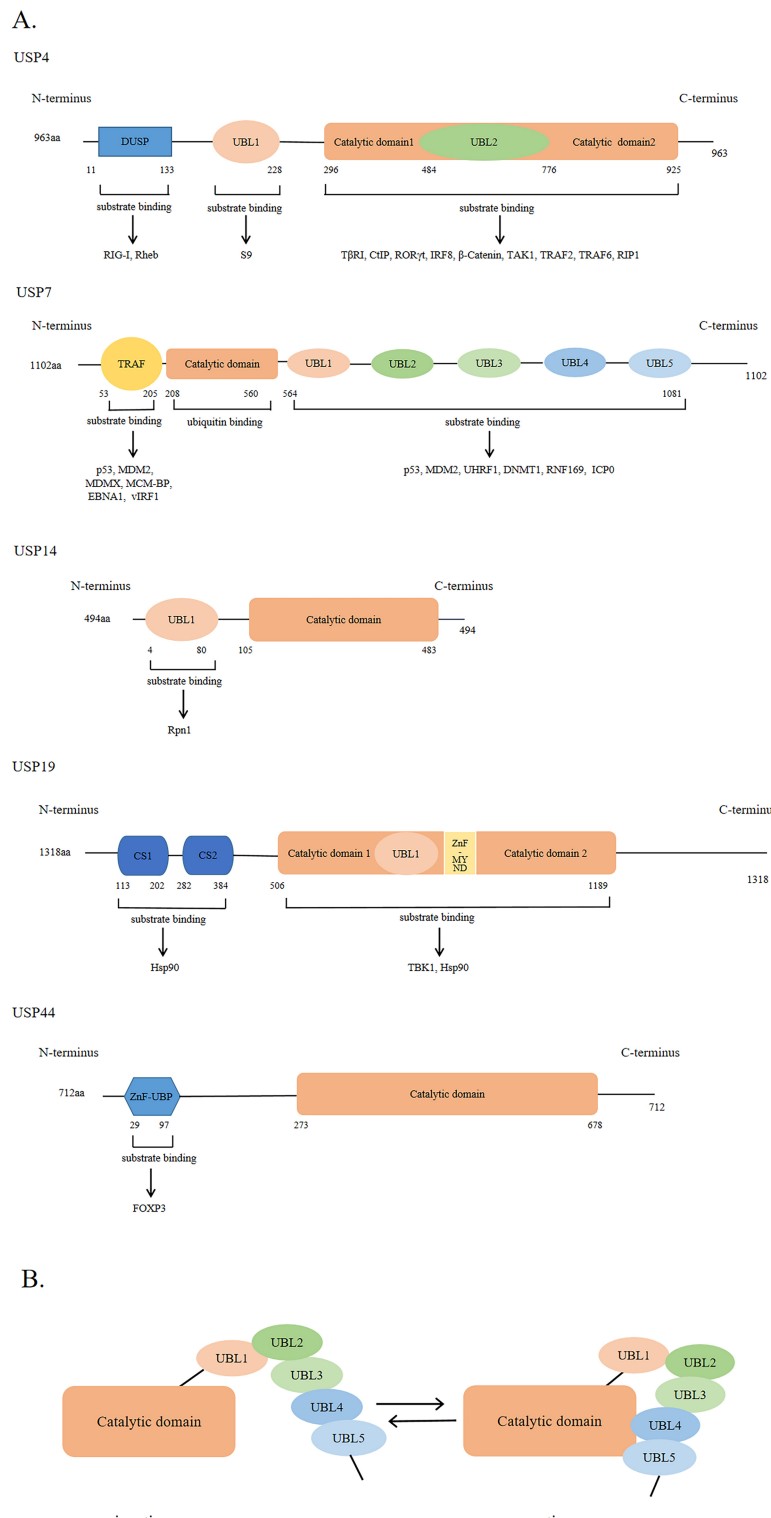

**Figure 2** **The general domain structure of USP4, USP7, USP14, USP19 and USP44.** (A) The DUSP (domain in USP) domain, UBL (ubiquitin-like) domain, TRAF (tumor necrosis (continued on next page...)

**Figure 2 (…continued)**
factor receptor associated factor)-like domain, catalytic domain, CS (CHORD and SGT1) domain, ZnF-MYND (zinc-finger myeloid translocation protein 8, Nervy and Deaf1) domain and ZnF-UBP (zinc-finger ubiquitin binding) domain are represented by different colors and shapes. The numbers indicate the amino acid residues. As shown by the arrow, different USP domains bind to different substrate proteins. (B) The five UBL domains of USP7 make up the UBL unit. UBL4 and UBL5 interact with the catalytic domain of USP7 to activate USP7; Conversely, inactivation.

repeat sequence (*Cohn et al., 2007*). This binding significantly increases the enzymatic activity of USP1 (*Cohn et al., 2007*; *García-Santisteban et al., 2013*). The KDM4A (histone demethylase lysine-specific demethylase 4A) is upregulated in PCa, which is crucial for the survival and growth of PCa cells (*Cui et al., 2020*). USP1 can promote the growth and tumorigenesis of PCa cells by stabilizing the expression of KDM4A (*Cui et al., 2020*). The SIX1 (sine oculis homeobox homolog 1) is a poor prognostic indicator in PCa. USP1 can interact with SIX1 and stabilize its protein levels in a GRP75 (glucose regulated protein 75)-dependent manner, thereby promoting the cellular proliferation and castration resistance of PCa (*Liao et al., 2021a*; *Liao et al., 2022*).

### USP2

USP2 is an oncoprotein overexpressed in PCa tissues (*Priolo et al., 2006*). USP2a is the largest isoform of USP2 (*Renatus et al., 2006*). *Priolo et al. (2006)* found that USP2a could protect the PCa cells from apoptosis, and its overexpression showed a tumorigenic effect in mice. In addition, the overexpression of USP2a can also enhance the proliferation and aggressiveness of PCa cells (*Benassi et al., 2012*). Studies have shown that the mRNA and protein expression levels of USP2a are upregulated in hepatocellular carcinoma (*Xiong et al., 2021*). USP2a can significantly promote the growth, invasion, and metastasis of liver cancer cells and is positively correlated with poor pathological grading, lymphatic metastasis, and poor prognosis (*Xiong et al., 2021*). Therefore, it can be speculated that USP2a might be a potentially effective target for cancer therapy. The upregulation of USP2 increases the levels of deubiquitinated substrates. For example, USP2 can stabilize the protein levels of FAS (fatty acid synthase), MDM2 (murine double minute 2), cyclin D1, and Aurora-A (*Chuang et al., 2018*). These proteins might play a critical role in the development and progression of PCa. Myc is an oncoprotein; its overexpression contributes to the development of PCa. USP2a can indirectly upregulate the expression of Myc to regulate the MDM2-p53-miR-34b/c axis by promoting the progression of cancer (*Benassi et al., 2012*; *Benassi et al., 2013*; *Nelson, De Marzo & Yegnasubramanian, 2012*). Fatty acid synthase (FAS) is overexpressed in prostate cancer, and it can protect the cancer cells from apoptosis (*Graner et al., 2004*). USP2a can promote the development of PCa by stabilizing FAS and preventing its proteasomal degradation. Seelan et al. suggested that USP2 could deubiquitinate acid ceramidase (ACDase) in the LNcap cell, and the protein expression levels of ACDase were highly correlated with USP2. Studies also showed that the mRNA and protein expression levels of ACDase were upregulated in the PCa cells (*Seelan et al., 2000*). The overexpression of ACDase could promote the proliferation of PCa cells *in vitro* and *in vivo* (*Mizutani et al., 2015*).

### USP4

Studies have shown that USP4 has lower protein and mRNA expression levels in PCa tissues as compared to those in normal tissues (*Chen et al., 2021b*). Therefore, it can be speculated that USP4 might be a potentially effective target for the treatment of PCa. In PCa, METTL3 (methyltransferase like 3) is an upstream negative regulatory molecule of USP4. The upregulation of METT3 levels leads to the downregulation of USP4 levels, thereby reducing its deubiquitination ability, which causes a reduction in the levels of ELAVL1 (ELAV like RNA-binding protein 1). Consequently, the reduction in the ELAVL1 levels increases the expression levels of ARHGDIA (Rho GDP dissociation inhibitor $\alpha$), which promotes the migration and invasion of PCa cells. It has been shown that the mRNA and protein expression levels of ARHGDIA are upregulated in PCa cells. In addition, the survival curve analysis showed a much shorter relapse-free survival of the PCa patients with higher *ARHGDIA* mRNA expression levels (*Chen et al., 2021b*). Therefore, it can be concluded that the invasion and metastasis of PCa cells can be inhibited by targeting and increasing the expression level of USP4.

### USP7

USP7 is also known as herpesvirus-associated ubiquitin-specific protease (HAUSP) (*Pozhidaeva et al., 2015*). It has been overexpressed in PCa cells and is correlated with the invasion of their cells (*Song et al., 2008*). It can affect the migration, invasion, and sphere formation in PCa cells (*Lee, Park & Kim, 2020*). The knockdown of the *USP7* gene can reduce the proliferation of PCa cells and induce their apoptosis (*Shin et al., 2020*). The PCa patients with high expression of *USP7* had a worse prognosis in a large clinical population (*Park et al., 2021*). USP7 can interact with multiple substrate proteins, such as AR, FOXA1, FOXO4, PTEN, PLK1, MDM2, and CCDC6, to stabilize their protein levels (*Chen et al., 2015*; *Morra et al., 2017*; *Park et al., 2021*; *Peng et al., 2019*; *Song et al., 2008*; *Van der Horst et al., 2006*; *Weinstock et al., 2012*). USP7 is positively correlated with the PLK1 expression; both the USP7 and PLK1 have strong clinical relevance and are correlated with the poor prognosis of patients (*Shin et al., 2020*). The tissue expression levels of CCDC6 and USP7 can be used as markers to assess the treatment effects of advanced PCa, and their high expression predicts poor prognosis in patients with advanced PCa (*Morra et al., 2017*).

### USP9X

USP9X, a conserved DUB, has a downregulated expression level in PCa tissues. The depletion of USP9X promotes the proliferation, migration, and invading abilities of PCa cells; these invasion and migration abilities are greater than the proliferation ability. *Zhang et al. (2019)* showed that the downregulation of the USP9X expression was correlated with poor differentiation and local invasion of PCa. This indicated that the PCa with low *USP9X* expression was highly deteriorated and had a negative impact on patients. USP9X can stabilize the levels of various protein substrates, such as PBX1, ERG, and IRS-2 (*Furuta et al., 2018*; *Liu et al., 2019*; *Wang et al., 2016*). PBX1 can promote the proliferation of PCa cells and is highly expressed in PCa. Studies have shown that the increased levels of PBX1 often lead to chemoresistance, which can be overcome in PCa patients by targeting the

USP7-PBX1 axis (*Liu et al., 2019*). ERG is a major driver of PCa. Blocking the development of PCa by knocking down the USP9X leads to the ubiquitination and degradation of ERG protein (*Wang et al., 2014*).

### USP10

The expression level of USP10 is upregulated in PCa tissues as compared to benign prostate tissues. USP10 can promote the proliferation of PCa cells and play an oncogenic role. The high expression of USP10 is correlated with the poor prognosis of patients. USP10 can deubiquitinate and stabilize the protein levels of G3BP2 (GTPase-activating protein-binding protein 2) and inhibit p53 activity to promote AR signaling pathway (*Takayama et al., 2018*). G3BP2 is upregulated in PCa and might serve as an independent prognostic marker for PCa patients (*Ashikari et al., 2017*).

### USP12

USP12 is a novel biomarker for PCa. Its expression levels were upregulated in PCa and CRPCa. The upregulation of USP12 expression is correlated with the poor prognosis of PCa patients. *McClurg et al. (2018b)* found that the PCa patients with high expression levels of USP12 had significantly shorter overall survival and shorter recurrence time. USP12 is also closely associated with the levels of proliferation marker MCM2. Therefore, it can be inferred that USP12 can promote the proliferation of PCa cells. Mechanistically, USP12 can promote the development of PCa by stabilizing MDM2 protein levels and reducing p53 levels (*McClurg et al., 2018b*). USP12 is a novel regulator of AR by interacting with Uaf-1 and WDR20 to enzymatically perform the deubiquitination of AR, thereby enhancing its stability and transcriptional activity (*Burska et al., 2013*; *McClurg et al., 2014*). On the other hand, USP12 can also stabilize AR proteins and enhance their transcriptional activity by stabilizing the AKT (protein kinase B) dephosphorylases (PHLPP and PHLPPL) (*McClurg et al., 2014*). Stabilizing the AR protein and increasing its transcriptional activity can promote the proliferation and survival of PCa cells (*Burska et al., 2013*).

### USP14

USP14 promotes the proliferation of PCa cells and is closely associated with its progression (*Liao et al., 2017*). It can also promote the development of PCa by stabilizing AR protein and ATF2 (activating the transcription factor 2) (*Geng et al., 2020*; *Liao et al., 2017*). USP14 is a novel regulator of AR (*Liao et al., 2017*). AR protein is overexpressed in PCa and plays a key role in its growth and progression (*Liao et al., 2017*). *ATF2*, an oncogene of PCa, facilitates the proliferation of PCa cells (*Geng et al., 2020*).

### USP15

The Oncomine public data showed that the expression levels of *USP15* were significantly elevated in PCa tissues, showing its potential clinical significance in PCa (*Padmanabhan et al., 2018*). Studies showed reported that USP15 was highly expressed in various malignancies, including gastric cancer. Moreover, the upregulation of USP15 expression was positively correlated with the clinical features of gastric cancer, such as tumor size, depth of invasion, lymph node involvement, tumor-node-metastasis stage, perineural

invasion, and vascular invasion. *USP15* is also strongly associated with poor prognosis of the patient and might be an oncogene in gastric cancer (*Zhong et al., 2021*). Notably, *Fukushima et al. (2017)* suggested that USP15 could inhibit the proliferation of the PCa cell line (PC-3). Mechanistically, USP15 inhibits the proliferation of PCa cells by antagonizing the NEDD4-induced ubiquitination of IRS-2, which leads to the reduction of tyrosine phosphorylation of IRS-2, resulting in a decreased IGF signaling (*Fukushima et al., 2017*). The IGF signaling pathway and increase in IGF levels contribute to the proliferation of PCa cells (*Ahearn et al., 2018*).

### USP16

The expression level of USP16 is higher in PCa tissues as compared to those in the normal prostate tissues and is correlated with the poor prognosis of PCa patients. USP16 can promote the cellular proliferation and growth of PCa (*Ge et al., 2021*). c-Myc, an oncoprotein, is involved in cellular proliferation. USP16 can regulate the growth of PCa cells by stabilizing c-Myc (*Ge et al., 2021*).

### USP17

USP17 has higher mRNA and protein levels in PCa tissues as compared to those in normal prostate tissues. The PCa patients with lower expression levels of USP17 have higher overall survival (*Baohai, Shi & Yongqi, 2019*). The knockdown of the *USP17* gene can promote the production of ROS to induce apoptosis and decrease p65 phosphorylation, thereby inhibiting the proliferation, migration, and invasion of PCa cells as well as the progression of PCa (*Baohai, Shi & Yongqi, 2019*).

### USP19

USP19 could regulate the growth of DU145, PC-3, and 22RV1 PCa cell lines but did not affect that of the LNCaP PCa cell line (*Lu et al., 2011*). The knockdown of the *USP19* gene could inhibit the progression of PCa in two ways; first, by causing defects in cell cycle progression and delaying the progression from G0/G1 to S phase, and second, by accumulating p27$^{Kip1}$ and inhibiting the ability of USP19 to regulate cell growth (*Lu et al., 2011*).

### USP22

USP22 has an important role in promoting malignant phenotypes by enhancing the stability of multiple cancer-associated protein targets through deubiquitination (*Schrecengost et al., 2014*). The mRNA and protein expression levels of USP22 are significantly upregulated in PCa. *USP22* is a pro-oncogenic factor in PCa. Animal studies showed that USP22 conferred an excessive proliferation phenotype *in vivo* (*McCann et al., 2020*). USP22 is positively associated with the progression of PCa. Its upregulation not only shortens the survival times of patients but also correlates with the recurrence and poor prognosis of patients (*Nag & Dutta, 2020*). Therefore, USP22 can be used as a diagnostic biomarker. Mechanistically, USP22 promotes the progression of PCa by stabilizing the AR protein levels and regulating the AR-Myc signaling pathway (*Schrecengost et al., 2014*).

### USP25

Although the clinical data showing the exact role of USP25 in PCa is currently lacking, *USP25* expression is downregulated in PCa tissues as compared to normal tissues (*Chen et al., 2021b*). The wnt/$\beta$-catenin pathway is an important pathway for the development and progression of PCa (*Cheng et al., 2019a*). Tankyrase (TNKS) is a key mediator of the Wnt signaling pathway. USP25 can stabilize TNKS protein to promote the proliferation of PCa cells (*Cheng et al., 2019a*).

### USP26

The expression level of USP26 is low in PCa tissues as compared to normal prostate tissues (*Wosnitzer et al., 2014*). A study reported that the mutations in the *USP26* gene were not involved in the development of PCa, suggesting that the role of USP26 in the development of PCa might be due to changes in its protein expression levels (*Dirac & Bernards, 2010*). USP26 is an AR signaling regulator, which reverses AR ubiquitination, regulates the AR transcriptional activity, and protects the AR proteins from degradation (*Dirac & Bernards, 2010*). USP26 has a cancer promoter role due to its regulatory effects on AR.

### USP33

The mRNA and protein expression levels of USP33, also known as VDU1, are upregulated in PCa (*Ding et al., 2021*; *Guo et al., 2020*). *Guo et al. (2020)* found that USP33 was an important tumor-promoting factor in PCa. It can inhibit apoptosis and enhance the cell survival of PCa cells. Current studies have shown that USP33 is an effective target for the treatment of PCa. USP33 promotes the development of PCa through two pathways. First, USP33 interacts with DUSP1 (dual-specificity phosphatase-1) to negatively regulate the JNK (cJUN NH2-terminal kinase) activation, thereby inhibiting the apoptosis of PCa cells (*Guo et al., 2020*). Second, PCa cells have upregulated levels of circ_ 0057558, which is a new biomarker for PCa and promotes cancer development by regulating the miR-206–USP33–c-Myc axis. Importantly, the expression levels of *USP33* and *circ _ 0057558* are positively correlated with PCa (*Ding et al., 2021*).

### USP34

*USP34* gene is correlated with the recurrence of PCa and is a predictive marker for its recurrence after radical prostatectomy. The high mRNA expression level of *USP34* has a protective effect on the recurrence of PCa. Studies have shown that the Th2 (type 2 T helper) and Tcm (central memory T cell) immune cells are associated with the recurrence of PCa after radical prostatectomy and are the independent recurrence prevention factors. *NDUFA13*, *UQCR11*, and *USP34* are involved in the infiltration of Th2 and Tcm cells in tumor tissues. The high expression of the *USP34* can activate the Th2 and Tcm immune cells and recognize tumor cells through cellular and humoral immune effector mechanisms, thereby delaying the recurrence of PCa and improving prognosis. Conversely, the high expression levels of *NDUFA13* and *UQCR11* can inhibit tumor recognition by the human immune system, thereby significantly increasing the probability of tumor recurrence (*Rui et al., 2019*).

### USP39

PCa cells, especially the AR-negative PCa cells have upregulated expression levels of *USP39* (*Huang et al., 2016*). USP39 is a biomarker, which predicts poor prognosis and recurrence in patients. Studies have shown that USP39 can affect the colony formation of cancer cells and tumor growth. The inhibition of USP39 can induce G2/M phase arrest and promote apoptosis (*Huang et al., 2016*). USP39 is positively correlated with EGFR (epidermal growth factor receptor) levels; USP39 upregulates the mRNA and protein expression levels of EGFR (*Huang et al., 2016*). Studies have shown that EGFR is involved in the bone metastasis of PCa (*Thoma, 2017*). Therefore, inhibiting the EGFR signaling pathway might suppress the metastasis of PCa (*Xiong et al., 2020*). It is speculated that USP39 is a potential target for the treatment of PCa.

### USP44

USP44 has a higher protein expression level in malignant and mPCa cells as compared to that in benign and weak mPCa cells (*Park et al., 2019*). USP44 can promote tumor formation and CSC (cancer stem cell) -like properties of PCa cells. USP44 could promote the invasive and migratory abilities of PCa cells *in vitro*, indicating its oncogenic behavior (*Park et al., 2019*). Mechanistically, USP44 can deubiquitinate to stabilize the EZH2 (enhancer of zeste homolog 2) protein levels and promote the development of PCa (*Park et al., 2019*). Moreover, EZH2 could facilitate the proliferation of PCa cells *in vitro* and *in vivo*. The overexpression of EZH2 is correlated with the deterioration of hormone-refractory mPCa. EZH2 is also a risk marker of the lethal progression of PCa (*Varambally et al., 2002*). Taken together, USP44 plays a role in the progression of PCa by regulating EZH2.

### USP46

USP46 has highly similar to USP12 and has overlapping biological functions in PCa (*McClurg et al., 2018a*). The increased mRNA expression level of *USP46* could predict a shorter relapse-free survival of patients. *McClurg et al. (2018a)* found that cancer development could be inhibited by targeting USP46. Moreover, USP46 can affect the progression of PCa by regulating the AR-AKT-MDM2-p53 signaling pathway (*Li et al., 2013*; *McClurg et al., 2018a*).

The correlations between USPs and PCa are summarized in Table 1.

## CONCLUSIONS

Despite the developments in the treatment of PCa, there is an increasing need for new therapeutic targets. DUBs play an important role in the development of PCa and are promising drug targets. The development of small molecule inhibitors, targeting DUBs, holds great promise for the treatment of PCa. Some inhibitors have been applied in clinical practices, such as b-AP15, ML323, and Spautin-1. However, these studies are very limited and require further investigation. In addition, USPs are correlated with numerous cancer types. Thus, developing effective inhibitors against USPs are a promising method of finding new drugs for the treatment of cancers, including PCa. This review summarized the correlations between USPs and PCa. It was speculated that DUBs, especially USPs,

**Table 1** Ubiquitin-specific peptidases (USPs) related to prostate cancer.

| USPs | Expression in prostate cancer | Pathways/Mechanism | Targets | Inhibitors | Reference |
|---|---|---|---|---|---|
| USP1 | high expression (mRNA) | stabilization of KDM4A protein levels regulation of GRP75-SIX1 signaling axis | KDM4A SIX1 | ML323 pimozide | *Liao et al. (2021a)*; *Ma et al. (2019)*; *García-Santisteban et al. (2013)*; *Cui et al. (2020)* |
| USP2 | high expression | regulation of MDM2-p53-Myc signaling axis stabilization of FAS protein levels stabilization of ACDase protein levels | MDM2 FAS ACDase | β-lapachone ML364 | *Priolo et al. (2006)*; *Nelson, De Marzo & Yegnasubramanian (2012)*; *Graner et al. (2004)*; *Seelan et al. (2000)*; *Gopinath et al. (2016)*; *Zhang, Zhao & Sun (2021)* |
| USP4 | low expression | regulation of ELAVL1-ARHGDIA signaling axis | ELAVL1 | Vialinin A | *Chen et al. (2021b)*; *Xu et al. (2021)* |
| USP7 | high expression | stabilization of PLK1 protein levels regulation of MDM2/MDMX-p53 signaling pathway regulation of AR signaling pathway regulation of localization of PTEN | PTEN AR KDM6A MDM2M DMX p53 PLK1 | P5091 P22077 Almac4 HBX19818 GNE-6776 Quinazolin-4(3H)-one HBX41108 | *Song et al. (2008)*; *Shin et al. (2020)*; *Peng et al. (2019)*; *Chen et al. (2015)*; *Li et al. (2002)*; *Qi et al. (2020)*; *Wang et al. (2021)*; *Dai et al. (2020)*; *Li et al. (2021)*; *Tang et al. (2017)* |
| USP9X | low expression | regulation of ERG signaling pathway regulation of MMP9-DRP1 signaling axis | ITCH ERG PBX1 | WP1130 G9 | *Zhang et al. (2019)*; *Lu et al. (2019)*; *Pal et al. (2018)* |
| USP10 | high expression | regulation of AR singaling pathway | G3BP2 | Spautin-1 | *Takayama et al. (2018)*; *Liao et al. (2019)* |
| USP12 | high expression | regulation of AR signaling pathway regulation of TP53-MDM2-AR-AKT signaling axis regulation of AKT pathway | H2A H2B AR MDM2 PHLPPs | GW7647 | *McClurg et al. (2018b)*; *McClurg et al. (2014)* |
| USP14 | unknown | stabilization of AR and MDM2 protein levels stabilization of ATF2 protein levels regulation of AR signaling | AR MDM2 ATF2 | IU1 S5 b-AP15 | *Liao et al. (2017)*; *Geng et al. (2020)*; *Xu et al. (2020)*; *Ming et al. (2021)* |
| USP15 | High expression (mRNA) | antagonizing the effect of NEDD4 to regulate IGF signaling pathway | Nedd4 | Mitoxantrone USP15-Inh | *Padmanabhan et al. (2018)*; *Fukushima et al. (2017)*; *Ward et al. (2018)*; *Niederkorn et al. (2022)* |
| USP16 | high expression | stabilization of c-Myc protein levels | c-Myc | unknown | *Ge et al. (2021)* |
| USP17 | high expression | regulation of EMT signaling pathway inhibition of USP17 can promote ROS production and inhibit NF-kB and p56 | EMT | unknown | *Baohai, Shi & Yongqi (2019)* |

**Table 1** (*continued*)

| USPs | Expression in prostate cancer | Pathways/Mechanism | Targets | Inhibitors | Reference |
|---|---|---|---|---|---|
| USP19 | unknown | regulation of p27 $^{Kip1}$ level | unknown | unknown | *Lu et al. (2011)* |
| USP22 | high expression | regulation of AR and c-Myc dual signaling pathway | AR<br>AR-v7<br>c-Myc | unknown | *Schrecengost et al. (2014)*;<br>*McCann et al. (2020)* |
| USP25 | low expression (mRNA) | regulation of Wnt signaling pathway | TNKS | unknown | *Chen et al. (2021b)*;<br>*Cheng et al. (2019a)* |
| USP26 | low Expression | regulation of AR signaling pathway | AR | unknown | *Wosnitzer et al. (2014)*<br>*Dirac & Bernards (2010)* |
| USP33 | high expression | regulation of DUSP1-JNK singaling axis<br>stabilization of c-Myc protein levels | DUSP1<br>c-Myc | unknown | *Guo et al. (2020)*<br>*Ding et al. (2021)* |
| USP39 | high expression | regulation of EGFR signaling pathway | EGFR | unknown | *Huang et al. (2016)*;<br>*Thoma (2017)* |
| USP44 | high expression | stabilization of EZH2 protein levels | EZH2 | unknown | *Park et al. (2019)* |
| USP46 | unknown | regulation of AR-AKT-MDM2-p53 signaling pathway<br>stabilization of AR, MDM2 and PHLPPs protein levels | H2A<br>H2B<br>AR<br>MDM2<br>PHLPPs | unknown | *Li et al. (2013)*;<br>*McClurg et al. (2018a)* |

**Notes.**

USPs, ubiquitin-specific peptidases; KDM4A, histone demethylase lysine-specific demethylase 4A; GRP75, glucose regulated protein 75; SIX1, sine oculis homeobox homolog 1; MDM2, murine double minute 2; FAS, fatty acid synthase; ACDase, acid ceramidase; KDM6A, histone demethylase lysine-specific demethylase 6A/UTX; MDMX, murine double minute X; AR, androgen receptor; ERG, ETS-related gene; MMP9, matrix metalloproteinase 9; DRP1, dynamin-related protein 1; PBX1, Pre-B cell leukemia homeobox-1; G3BP2, GTPase-activating protein-binding protein 2; ATF2, transcription factor 2; IGF, insulin growth factor; EMT, epithelial-mesenchymal transition; DUSP1, dual-specificity phosphatase-1; JNK, cJUN NH2-terminal kinase; EGFR, epidermal growth factor receptor; EZH2, enhancer of zeste homolog 2.

might be effective therapeutic targets, provided that their molecular mechanism in PCa is extensively investigated. Therefore, to develop small molecule inhibitors against USPs for the clinical treatment of PCa and other cancers, more studies regarding the molecular mechanisms of USPs in various cancers and how these USPs themselves are regulated in future.

### Funding

This study was supported by the National Natural Science Foundation of China (81872005). The funders had no role in study design, data collection and analysis, decision to publish, or preparation of the manuscript.

### Grant Disclosures

The following grant information was disclosed by the authors:
The National Natural Science Foundation of China: 81872005.

### Competing Interests

The authors declare there are no competing interests.

## Author Contributions

- Yunfei Guo performed the experiments, analyzed the data, prepared figures and/or tables, authored or reviewed drafts of the article, and approved the final draft.
- Shuaishuai Cui analyzed the data, prepared figures and/or tables, and approved the final draft.
- Yuanyuan Chen analyzed the data, prepared figures and/or tables, and approved the final draft.
- Song Guo analyzed the data, prepared figures and/or tables, and approved the final draft.
- Dahu Chen conceived and designed the experiments, analyzed the data, prepared figures and/or tables, authored or reviewed drafts of the article, and approved the final draft.

## Data Availability

This literature review has no raw data or code.

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
