# Peer review of "Ubiquitin specific peptidases and prostate cancer"

_PeerJ, doi:10.7717/peerj.14799_

## Round 0.1 · original submission · Major Revisions

The manuscript's language and an improved literature overview need to be addressed. Please, also refer to the authors' instructions (for literature reviews) for some aid to enhance your manuscript.

Reviewer 1 ·

Basic reporting

This review summarizes the function of ubiquitin-specific proteases (USP) in prostate cancer development and progression. It is a comprehensive summary for the audience who is interested in understanding how USP regulates prostate cancer and researchers that intended to use USP as a therapeutic target for prostate cancer.

Experimental design

Resources are adequately cited and the previous research work covers all the USPs.

Validity of the findings

The conclusion in the review is quite general. Authors suggested more clinical studies and trials need to focus on developing inhibitors for prostate cancer treatment, and in order to develop inhibitors, the molecular mechanism of prostate cancer need to be extensively studied.

Reviewer 2 ·

Basic reporting

The language in the manuscript is clear, however, sentences formations have to be improved. The manuscript needs to be organized better. For example- there is not need for separate section entitled- Why this review is needed and who it is intended for (line 89). The information can be a part of introduction itself.
Lines 98-99 are not necessarily true- as there are several published reviews which discuss USPs and their role in cancer progression.

Experimental design

Survey Methodology section is not needed as a separate section for literature review.

It is not customary to thank cited researchers in the Acknowledgements section. Citing appropriate literature is adequate.

Apart from this - I would encourage the authors to revise the writing style to make it more professional to meet the journal requirements.

Is there any reason why the authors have a figure for USP7 but not for others? If not, schematics for other USPs structures where possible will increase the readers' experience.

Validity of the findings

I would advice the authors to limit the personal opinions. For instance- line 135-136 "Therefore, It may be a very promising therapeutic method for cancer treatment through targeting USPs."

Reviewer 3 ·

Basic reporting

The review “Ubiquitin specific peptidases and prostate cancer” by Yunfei Guo and colleagues summarized the functional roles of ubiquitin speciûc proteases in the development and progression of prostate cancer and explored the potential applications of using these ubiquitin speciûc proteases as therapeutic targets for prostate cancer.
In general, the review helps in understanding the roles of ubiquitin specific proteases in the development and progression of prostate cancer and designing of cancer drug targeting strategies. However, the manuscript can be considerably improved. I can recommend it for publishing with a few questions outlined.

Experimental design

1.Authors are suggested to correct the grammatical error.

2.Authors are suggested to cite the other reviews on targeting the deubiquitinases regulated pathways for prostate cancer therapeutics.

3.Correct the Line 37-38 “which is an 8.5 kDa protein by consisting of 76 amino acids that can be post-translationally coupled to a substrate protein, usually at a lysine residue” as “which is a 8.5 kDa protein, consisting of 76 amino acids that can be post-translationally coupled to a substrate protein, usually at a lysine residue”

4.Line 81, authors write “there are studies showing a strong link between the expression...........” but only one reference is given. Authors are suggested to add other references.

5.In section Why this review is needed and who it is intended for, references are missing. Authors are suggested to add references.

6. Line 98-99, authors claim that “However, no review summarizes the relationship of USPs with cancers, particularly prostate cancer”. but Md. Tariqul Islam et. al. In their review “Targeting the signalling pathways regulated by deubiquitinases for prostate cancer therapeutics, Cell Biochem Funct. 2019;37:304–319” showed the involvement of USPs in prostate cancer. Author are suggested to to mention their work in this article.

7.Line 119, correct the ASP as Asp.

8.Abbreviation (like UPS4, METTL3, USP7, ARHGDIA etc.) are in Italic format at many places so authors are suggested to correct them throughout the manuscript.

9.Authors are suggested to write the full form of FAS in line 171 rather than in line 173.
Line 174, correct the “deubiquitinated Acid ceramidase” as deubiquitinated acid ceramidase

10.Authors are suggested to add DOI in reference “Tobias JW, and Varshavsky A. 1991. Cloning and functional analysis of the ubiquitin-specific protease gene UBP1 of Saccharomyces cerevisiae. J Biol Chem 266:12021-12028”.

Validity of the findings

The article meets the argument stated in introduction. Some of the sections should written with more evidence from literature. The conclusion meets the claims stated but does not include the future directions, authors are suggested to mention in the conclusion.

---

## Round 0.2 · accepted · Accept

Thank you for choosing PeerJ and submitting your work on prostate cancer.

Reviewer 3 ·

Basic reporting

The revised review has corrections for grammatical error before and have clear language for readers to follow through the article. The introduction is improved and have been checked for any abbreviation errors.

Experimental design

The revised review follows scope of the journal and suggested improvements. The future directions have been updated as suggested. The abbreviations corrections have been done throughout the manuscript as well. The citations suggested were also included in the revised manuscript.

Validity of the findings

The revised manuscript does follow the journal policy and is well written now. The introduction is clear and lucid. The revised manuscript is now ready for publication.

Additional comments

The revised manuscript have been thoroughly corrected and meets the criteria as required.